# KVFlow: Efficient Prefix Caching for Accelerating LLM-Based Multi-Agent Workflows

Zaifeng Pan[1]    Ajjkumar Patel[1]    Yipeng Shen[1]    Zhengding Hu[1]*    Yue Guan[1]
Wan-Lu Li[1]    Lianhui Qin[1]    Yida Wang[2]    Yufei Ding[1]

[1] UCSD    [2] AWS

## Abstract

Large language model (LLM) based agentic workflows have become a popular paradigm for coordinating multiple specialized agents to solve complex tasks. To improve serving efficiency, existing LLM systems employ prefix caching to reuse key-value (KV) tensors corresponding to agents' fixed prompts, thereby avoiding redundant computation across repeated invocations. However, current systems typically evict KV caches using a Least Recently Used (LRU) policy, which fails to anticipate future agent usage and often discards KV caches shortly before their reuse. This leads to frequent cache misses and substantial recomputation or swapping overhead. We present KVFlow, a workflow-aware KV cache management framework tailored for agentic workloads. KVFlow abstracts the agent execution schedule as an Agent Step Graph and assigns each agent a steps-to-execution value that estimates its temporal proximity to future activation. These values guide a fine-grained eviction policy at the KV node level, allowing KVFlow to preserve entries likely to be reused and efficiently manage shared prefixes in tree-structured caches. Moreover, KVFlow introduces a fully overlapped KV prefetching mechanism, which proactively loads required tensors from CPU to GPU in background threads for agents scheduled in the next step, thereby avoiding cache miss stalls during generation. Compared to SGLang with hierarchical radix cache, KVFlow achieves up to $1.83\times$ speedup for single workflows with large prompts, and up to $2.19\times$ speedup for scenarios with many concurrent workflows.

## 1 Introduction

LLM-based agentic workflows coordinate multiple specialized agents, each defined by a fixed prompt and responsible for a specific subtask, to solve complex problems in a modular and interpretable way [1, 2, 3, 4, 5]. For example, MetaGPT [3] structures agent collaboration around software engineering roles such as Product Manager and Engineer. While this design improves reusability and coherence, it also leads to high inference latency due to the need to repeatedly invoke LLMs for each agent throughout the workflow.

To alleviate this overhead, existing agentic frameworks and applications [3, 6, 7, 8, 9, 10] rely on LLM serving systems [11, 12, 13] equipped with system-level optimizations. A prevalent technique is prefix caching [12, 14], which reuses the key-value (KV) tensors produced by self-attention layers for static prompt tokens across decoding steps and requests. This is particularly beneficial in agentic workflows, where each agent is initialized with a fixed prompt specifying its name, responsibilities, and behavioral traits. Since these prompts remain constant across iterations, prefix caching avoids redundant computation on static content and significantly reduces per-agent inference latency.

---

*Corresponding author

39th Conference on Neural Information Processing Systems (NeurIPS 2025).

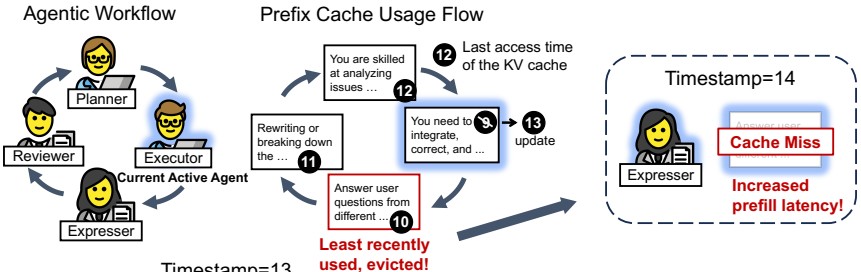

Figure 1: A cyclic agentic workflow abstraction consisting of four agents, Planner, Executor, Expresser, and Reviewer, adapted from [5]. At timestamp 13, the Executor is active and its KV cache is updated, which causes the Expresser's cache to be evicted due to the LRU policy. At timestamp 14, when the Expresser becomes active again, a cache miss occurs and results in increased prefill latency.

However, prefix caching alone is insufficient in the presence of limited GPU memory. Existing systems typically adopt a Least Recently Used (LRU) policy to evict KV caches that have not been accessed recently. We observe that this strategy can lead to suboptimal performance in agentic workflows. For instance, as illustrated in Figure 1, consider a workflow [5] where four agents are organized into a sequential execution pipeline that is invoked iteratively. During the execution of the Executor agent shown in the figure, the LRU policy identifies the Expresser's KV cache as the eviction candidate since it has not been accessed recently. This results in a cache miss when the workflow proceeds to the Expresser agent, despite its imminent reuse. Such eviction behavior introduces unnecessary recomputation and degrades the overall efficiency of agentic execution.

To address the limitations of existing LLM serving systems in agentic workflows, we present KVFlow, a workflow-aware KV cache management framework. We first introduce the *Agent Step Graph*, a flexible abstraction that captures execution dependencies among agents and supports a wide range of workflow structures, including conditional branching and synchronization barriers. Each agent node in the graph is associated with a computed *steps-to-execution* value, which estimates how soon the agent is expected to run. This value is derived through step aggregation functions that propagate across the graph, enabling KVFlow to reason about dynamic and structured execution patterns.

At runtime, KVFlow leverages this information to optimize cache behavior in two key ways. First, instead of using LRU, KVFlow adopts a workflow-aware eviction strategy that prioritizes evicting KV caches belonging to agents with large steps-to-execution. Since multiple agents can share common prefixes through a tree-structured cache, we further assign eviction priorities at the cache node level to enable fine-grained and efficient management. Second, KVFlow introduces a fully overlapped KV prefetching mechanism that proactively loads required KV tensors from CPU to GPU ahead of time, as we can predict the next invoked agents from the Agent Step Graph. This effectively eliminates prefix cache misses without stalling generation. Together, these optimizations significantly improve cache efficiency and reduce latency in executing agentic workflows.

In summary, this paper makes the following contributions:

- We identify a fundamental inefficiency in existing LLM serving systems, where the widely used LRU-based KV cache eviction strategy leads to suboptimal performance under agentic workflows.

- We propose KVFlow, a workflow-aware KV cache management optimization that prioritizes eviction based on agent execution order and eliminates cache miss overhead via fully overlapped prefetching.

- We conduct a comprehensive evaluation for KVFlow, showing that it significantly reduces cache miss overhead, achieving up to $1.83\times$ and $2.19\times$ speedups over SGLang with hierarchical radix cache under single workflows with large prompts and many concurrent workflows, respectively.

## 2 Background

**Prefix Caching in LLM Serving Systems.** To facilitate fine-grained prefix reuse and eliminate redundant storage, modern LLM serving systems [11, 12] organize the KV cache into a tree structure on the GPU, where each node stores a segment of tokens and its corresponding KV tensors. Upon receiving a new request, the system matches the prefix from the root of the tree and concatenates the KV tensors along the matched path to reconstruct the full cached prefix. When GPU memory becomes insufficient, the system evicts nodes based on an LRU policy. Memory exhaustion can arise for two reasons. One common scenario is a high volume of concurrent user requests, each executing different agentic workflows, which leads to a large number of active KV cache entries. Another scenario occurs when the agent prompts are very large while the hardware capacity is limited.

Additionally, CPU memory can be configured as a secondary cache layer to back up evicted KV tensors, allowing cache swapping over PCIe. Despite the PCIe latency, swapping remains significantly faster than recomputing the KV tensors [15, 16].

**Agentic Workflow.** An agentic workflow [8, 10, 9, 7] is an LLM application paradigm that structures multiple agents into an execution graph to collaboratively solve complex tasks. Compared to fully autonomous agents [17, 18, 19], agentic workflows leverage human domain expertise to achieve more consistent and robust performance across diverse tasks [3, 20, 21, 22, 23, 24, 25, 26]. The execution of each agent typically involves one or multiple LLM calls, with prompts composed of a *fixed* part and a task-specific *dynamic* part. The fixed part usually encodes the agent's role, behavioral instructions, task description, and few-shot learning examples, and can be substantially large. For example, the fixed prompts of the TestBench Agent and the RTL Generator Agent in [22] contain lengthy few-shot learning examples, with over 3000 and 1000 tokens, respectively. Consequently, caching the corresponding KV of the fixed parts can significantly reduce prefill latency and improve the overall workflow execution efficiency. In contrast, the dynamic parts often contain the input questions or instructions from users, which are less valuable for caching.

## 3 Design of KVFlow

In this section, we present the design of KVFlow, which enhances prefix cache management for agentic workflows through two key techniques. First, we introduce a workflow-aware eviction policy that prioritizes KV nodes based on future usage, improving over the default LRU strategy. Second, we propose an overlapped KV prefetching mechanism that hides CPU-GPU transfer latency via proactive loading and status-aware scheduling.

### 3.1 Workflow-Aware Eviction Policy

Existing LLM serving systems typically adopt an LRU eviction policy, which becomes suboptimal under agentic workflows. Specifically, an agent that is about to execute may have been idle for a long time, while an agent that has just completed its execution might not be needed again in the near future. Moreover, the suffixes dynamically generated by a recently executed agent often vary rapidly with task progress and are unlikely to be reused, yet they are still temporarily retained in the cache. With workflow information, we can predict the upcoming execution sequence of agents, enabling more informed eviction decisions and avoiding the inefficiencies caused by LRU.

**Agent Step Graph and Steps-to-Execution.** To make eviction decisions based on workflow structure, we first need to capture the dependency relationships among agents. However, agent interactions in real-world workflows are highly diverse. As illustrated in Figure 2(a), the two workflows differ significantly: in the upper example, the Expresser agent depends on both Executor1 and Executor2; in contrast, the lower workflow contains conditional branches, where Expresser can be triggered after either executor completes. Traditional abstractions such as control-flow graphs (CFGs) or DAGs are insufficient to uniformly capture such diverse execution semantics.

To address this, we introduce the *Agent Step Graph* abstraction, where each node corresponds to an agent invocation and edges encode dependency relations. Unlike conventional graphs, each node in the Agent Step Graph is associated with a *step aggregation function* that determines how its

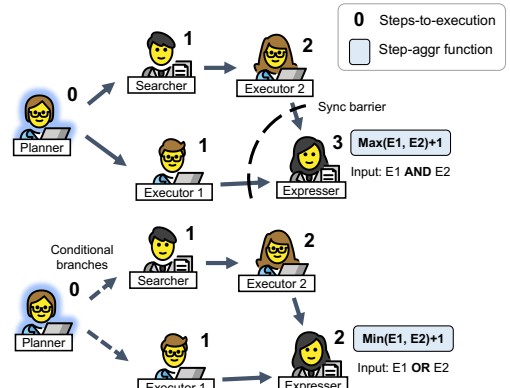

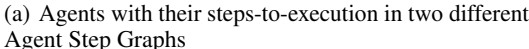

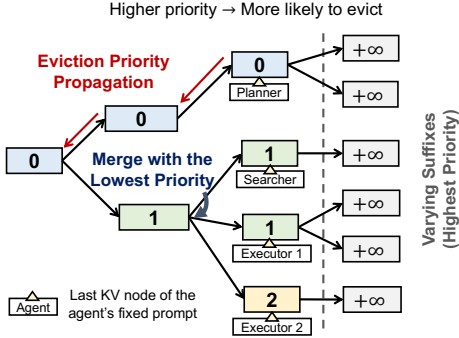

(a) Agents with their steps-to-execution in two different Agent Step Graphs

(b) Eviction priority assignment for each KV node within a cache tree, where multiple agents can share partial prefix prompts.

Figure 2: Illustration of the workflow-aware eviction policy. (a) Each agentic workflow is abstracted as an Agent Step Graph, where steps-to-execution values are computed using step aggregation functions over dependency edges. (b) These values are propagated through the cache tree to assign eviction priorities at the KV node level. Nodes with smaller steps-to-execution are retained longer, reducing the chance of premature eviction.

steps-to-execution is derived from its predecessors. For prefix cache management, we focus solely on the earliest possible execution step of each agent and abstract away the specific type of dependency.

For example, in the upper workflow of Figure 2(a), the Expresser agent requires both upstream executors to complete, so its step value is computed as $\max(E1, E2) + 1$. In the lower workflow, either path suffices, so the step value becomes $\min(E1, E2) + 1$. By applying these aggregation functions recursively, the Agent Step Graph enables unified computation of steps-to-execution across arbitrary multi-agent workflows.

**Workflow-Aware Eviction Priority Assignment.** Based on the steps-to-execution in the Agent Step Graph, we design a fine-grained eviction strategy that assigns priorities to KV cache nodes. As illustrated in Figure 2(b), agents with larger steps-to-execution are more likely to be evicted. Importantly, since agents may share common prefix segments in a tree-structured cache layout, we assign eviction priorities at the cache node level rather than at the agent level.

Specifically, we assign priorities only to the fixed prompt portion of each agent; all varying suffixes are always given the highest eviction priority to facilitate early eviction. For each agent, its steps-to-execution value is assigned to the last node of its fixed prompt and propagated upwards through the tree. When a node aggregates inputs from multiple agents, we assign it the minimum (i.e., least evictable) priority among its children, ensuring that shared nodes are retained as long as they are useful to any agent in the near future.

This propagation scheme yields a priority map over the cache tree that dynamically reflects workflow-driven reuse potential. When GPU memory becomes constrained, KVFlow first evicts varying suffixes, and then incrementally evicts prefix KV nodes in descending order of their assigned priority, favoring eviction of those unlikely to be reused soon. This design naturally accommodates multiple concurrent workflows, with conflicts at shared nodes resolved by choosing the lowest (most conservative) priority across workflows.

**Pseudocode.** To more clearly illustrate how KVFlow integrates with KV cache management, we include pseudocode for both eviction priority assignment and the eviction procedure. When a new agent request arrives with its Agent Step Graph (ASG) information, KVFlow updates the eviction priority of each cache node following Algorithm 1. When the system needs to evict cache nodes on the GPU to free memory (e.g., during prefill, decode, or prefetch), it proceeds from the leaf nodes in the cache tree based on their priorities, as shown in Algorithm 2.

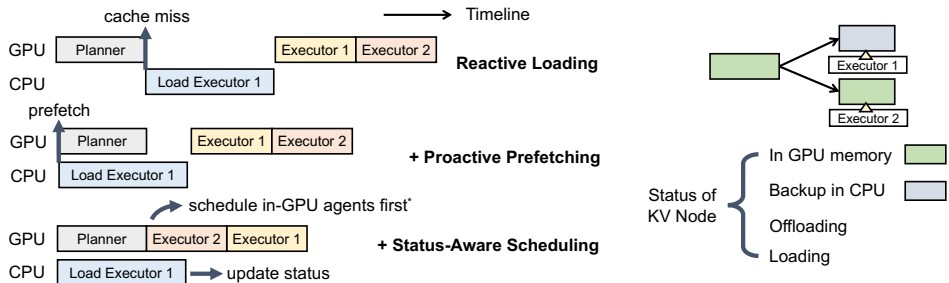

Figure 3: Illustration of overlapped KV prefetching. Compared to reactive loading, KVFlow combines proactive prefetching that loads upcoming agents in advance with status-aware scheduling, minimizing the CPU-GPU transfer overhead. *The in-GPU agents can be within the same workflow or from another concurrent one.

---

**Algorithm 1:** Priority Assignment

**Function** `PriorityAssign`(*ASG*)
    **for** *step*, *agent* in *ASG*
        *node* = *agent*.last_fixed_prompt_node;
        **while** *node* != *root*
            *node*.counter[*step*] += 1;
            *node*.priority = min(*node*.priority, *step*);
            *node* = *node*.parent;

**Algorithm 2:** Eviction Procedure

**Function** `Evict`(*tree_cache, required*)
    *leaves* = get_leaf_nodes(*tree_cache*);
    heapify(*leaves*); `// Construct max heap based on priority`
    **while** *free_gpu_memory* < *required*
        *node* = heappop(*leaves*);
        *free_gpu_memory* += evict(*node*);
        **if** *node*.parent becomes a leaf
            heappush(*leaves*, *node*.parent);

---

## 3.2 Overlapped KV Prefetching

While the workflow-aware eviction strategy avoids prematurely evicting agents that are about to execute, cache misses can still occur when an agent needs to run again after its KV cache has been evicted. This is particularly costly for long prompts, as recomputing the KV cache from scratch incurs significant overhead. To mitigate this, we treat CPU memory as a secondary cache for storing the fixed prompt KV of evicted agents.

When CPU caching is available, existing systems typically adopt a *reactive loading* strategy, as illustrated in the top timeline of Figure 3. For instance, if Executor 1's prefix cache has been offloaded to CPU memory, the system reactively loads it back only when Executor 1 is scheduled, thereby avoiding recomputation. However, CPU-to-GPU data transfers still introduce noticeable latency, especially for long prefixes.

**Proactive Prefetching.** To reduce this transfer overhead, we propose a *proactive prefetching* mechanism that leverages workflow information to asynchronously load the required KV cache in advance. As shown in the second timeline of Figure 3, while Planner is executing, the system anticipates that Executor 1 will be invoked next and proactively prefetches its KV cache from CPU to GPU. Since the execution of agents primarily involves model forward on the GPU and next token sampling (with model outputs transferred from the GPU to the CPU), and KV loading involves CPU-to-GPU transfer, the two operations utilize different hardware resources and can proceed in parallel without interference. Notably, PCIe enables full-duplex transfers, allowing bidirectional communication between CPU and GPU without contention. When the workflow contains branching, the system conservatively prefetches all agents that may be executed next based on the Step Graph, within a predefined limit on the number of concurrent prefetches.

However, prefetching alone is not always sufficient. When the current agent's execution time is shorter than the prefetch duration, generation may still be blocked by incomplete KV loading. This is common in high-concurrency settings, where multiple workflows compete for CPU-GPU bandwidth

and cause queuing delays. This scenario is depicted in the second timeline of Figure 3, where Executor 1's generation is stalled despite prefetching.

**Status-Aware Scheduling.** To further reduce GPU idle time, we enhance the request scheduling policy with status awareness. In each scheduling step, if a request's prefix cache is still in the loading process, the scheduler temporarily skips it and prioritizes other ready requests, such as Executor 2 in Figure 3 or those from other concurrent workflows. To support this, we associate each cache node with a status variable, which can be one of four states: *in GPU memory*, *backup in CPU*, *loading*, or *offloading*. The scheduler inspects all nodes required by a request, skips any with *loading* status to avoid redundant load attempts, and only dispatches the request once all dependencies are available. Upon completion, the background load thread updates the status of the cache nodes, informing readiness to the scheduler. Similarly, nodes in the *offloading* state are excluded from eviction decisions to avoid race conditions during memory reclaiming.

As illustrated in the third timeline of Figure 3, by combining proactive prefetching with status-aware scheduling, KVFlow effectively eliminates cache misses and fully overlaps GPU computation with prefetching, thereby hiding the CPU-GPU transfer latency.

## 3.3 Implementation

We implement the prototype of KVFlow based on SGLang v0.4.4 [12], an efficient LLM serving system that provides both a backend for LLM execution and a frontend interface for application development. SGLang's backend manages the prefix KV cache using a radix tree. We extend this mechanism to support our workflow-aware eviction policy and fully overlapped KV prefetching. In addition, we modify both the frontend and backend of SGLang to support the transmission of agentic workflow information.

While our current prototype is integrated into SGLang's frontend API, our method is not limited to SGLang. It can be adapted to other agentic workflow frameworks by modifying the HTTP requests that the frontend sends to the server. Our backend optimization can also generalize to other LLM serving systems. For instance, vLLM [11] internally adopts automatic prefix caching and applies LRU at the block level. Our workflow-aware eviction logic can be applied at this block granularity by maintaining a priority score for each block, enabling anticipatory eviction and prefetching.

**Step Information Capture.** Capturing the steps-to-execution information generated from the Agent Step Graph is essential for guiding our optimizations at runtime. In our implementation, we assume that each `sgl.function` corresponds to an independent agent. During execution, we perform a just-in-time substitution of the LLM call to embed workflow metadata into the HTTP request. This metadata includes the identity of the current agent and the steps-to-execution of all agents within the Agent Step Graph, indicating which agents may be invoked in subsequent steps. Upon receiving this information, the backend can update eviction priorities in the KV cache tree accordingly and trigger prefetching if the evictable GPU memory is large enough.

**Prompt Segmentation.** Besides capturing the step graph topology, we also need to track the last KV nodes of each agent's fixed prompt, as shown in Figure 2. Distinguishing the fixed and dynamic parts of the prompt within a request presents a challenge. We offer two alternative solutions. First, we introduce a primitive interface that allows users to explicitly mark the end position of the fixed part. Second, we design a heuristic approach that tracks the agent's cache hit history and treats the consistently hit prefix as the fixed part. To avoid stale entries, KVFlow supports an adaptive mechanism that automatically removes fixed prompt nodes if they have not been reused beyond a threshold period.

**Client Tracking.** In serving scenarios, multiple agentic workflows may execute concurrently on the same backend instance. Existing serving systems do not distinguish between request sources, potentially leading to naming conflicts. For example, two different workflows might both define an agent named "Planner". To address this, we assign a unique client ID to each application. The client ID is attached to every request sent to the backend, allowing the system to disambiguate agent identities and avoid interference between workflows originating from different clients.

## 3.4 Limitations

Our primary focus is on structured agentic workflows where future agent invocations can be estimated. KVFlow can handle dynamic workflows as long as the execution order of agents within a certain future window can be predicted at the current step. However, in highly unpredictable workflows where future execution cannot be inferred, KVFlow does not offer performance gains. In such cases, KVFlow gracefully falls back to SGLang's default behavior by setting equal priority for all agent cache and disabling prefetching, without introducing additional overhead or correctness issues.

# 4 Evaluation

We evaluate KVFlow across a range of microbenchmarks to understand its performance under different caching and execution conditions. Our experiments aim to answer the following key questions: (1) Can KVFlow reduce end-to-end latency for individual workflows with large prompt prefixes and limited GPU memory? (2) How does KVFlow perform under high concurrency, where multiple workflows run in parallel? To answer these questions, we first analyze single-workflow latency in Section 4.1, and then study multi-workflow execution in Section 4.2.

Since KVFlow only modifies system-level cache management without affecting the model weights, prompts, or decoding logic, it is guaranteed to preserve the semantic correctness of the output. Therefore, our evaluation focuses exclusively on system performance metrics like the latency.

## 4.1 Single-Workflow Latency

We begin by evaluating the latency of executing a single agentic workflow under batch size = 1. This single-request latency reflects interactive usage scenarios where workflows are triggered individually by a user, such as in notebooks or development tools. Unlike online serving systems that rely on batching for throughput, these settings prioritize responsiveness for individual requests.

We benchmark a sequential 10-stage agentic workflow. As described in Section 2, each agent's input prompt consists of a fixed prefix (shared across invocations) and a dynamic suffix (which varies across runs). We generate synthetic input prompts by randomly sampling token sequences with controlled lengths for both parts. We evaluate two variants: (a) fully deterministic sequential workflows where each stage only has one agent, i.e., branches=1; and (b) moderately dynamic workflows, where each stage randomly selects one of two agents with partially shared prefixes, i.e., branches=2. The latter one introduces moderate unpredictability while still preserves opportunities for prefetching.

**Models and testbeds.** We conduct experiments on Qwen2.5-32B on an NVIDIA H100 GPU with 80GB memory and 64 GB/s PCIe Gen5 bandwidth. Qwen uses 40 attention heads and 8 KV heads. We adopt deterministic decoding (temperature = 0, greedy sampling) to ensure consistent latency measurements. This setting represents scenarios with tight GPU memory constraints when long fixed prefixes contend for cache space.

**Baselines.** We compare KVFlow against two SGLang configurations. The first, denoted as **SGLang**, maintains a radix-structured KV cache in GPU memory without CPU backup. When GPU memory is insufficient, prefix nodes are evicted and must be recomputed from scratch upon reuse. For the second configuration, denoted as **SGLang w/ HiCache**, we enable the hierarchical radix cache in SGLang, which is SGLang's default CPU-based cache extension. It extends SGLang's radix tree by asynchronously backing up frequently used cache nodes to host memory. If a node is accessed after eviction, it is loaded back from the CPU instead of being recomputed. To further reduce CPU-GPU transfer cost, SGLang with HiCache overlaps the GPU computation of layer $l$ with the loading of layer $l+1$, forming a simple two-stage pipeline. Both of SGLang and SGLang w/ HiCache use LRU eviction policy when GPU memory is used up. We also include vLLM [11] as a baseline, which organizes KV cache into blocks and adopts an LRU-based block eviction policy.

**Evaluation Methods.** We first warm up the cache by executing each agent's fixed prompt multiple times, ensuring its prefix cache is constructed and backed up to CPU memory (for HiCache). We then execute the 10-stage workflow ten times, each with a varying dynamic suffix, to obtain the end-to-end latency for these ten runs. This simulates realistic usage patterns with repeated workflow invocations

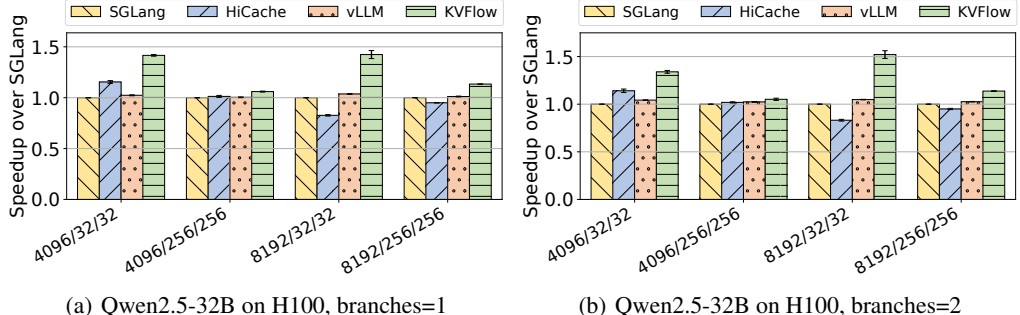

(a) Qwen2.5-32B on H100, branches=1      (b) Qwen2.5-32B on H100, branches=2

Figure 4: Speedup over SGLang (GPU-only cache) on Qwen2.5-32B for a 10-stage workflow. Left: branches=1 (deterministic sequential). Right: branches=2 (each stage randomly selects one of two agents). Horizontal axis: *Fixed part token / Dynamic part token / Output token*.

or loop-like behavior [5]. We then repeat this process ten times to report the average latency and the standard deviation.

**Results.** Figure 4 shows the speedup over SGLang (GPU-only cache) with error bars under different prompt configurations *Fixed / Dynamic / Output*. We intentionally test large fixed prefix sizes (e.g., 8192 tokens) to exceed GPU memory and force evictions.

Across all settings, KVFlow consistently achieves the highest speedup, demonstrating its effectiveness in both fully deterministic workflows (branches=1) and moderately dynamic workflows (branches=2). For instance, under 4096/32/32 with branches=1, KVFlow outperforms SGLang w/ HiCache by $1.24\times$, and the GPU-only SGLang baseline by $1.42\times$. This is because our workflow-aware eviction and prefetch strategy effectively hides the CPU-GPU transfer overhead. While HiCache reduces recomputation overhead, it still suffers from pipeline cold-start and limited overlap when compute is shorter than transfer.

We also observe that SGLang w/ HiCache generally performs better than the GPU-only SGLang baseline, as loading from CPU is typically faster than recomputing. However, in some large-context settings on H100 (e.g., 8192/32/32), HiCache shows marginal or degraded performance. This may stem from suboptimal scheduling in SGLang's pipelining logic, where CPU-GPU transfer is not effectively overlapped under memory contention or high transfer volume.

As the number of output tokens increases, the relative gain from KVFlow diminishes. The reason is that in these settings, the LLM decoding latency dominates total runtime, reducing the proportion of time affected by cache loading. There are many works optimizing the time-consuming decoding steps for the auto-regressive LLMs, including speculative decoding [27, 28, 29], KV cache sparsity [30], and early exit [31], which are orthogonal to our work and can be co-applied with KVFlow.

Meanwhile, the speedup from KVFlow increases with fixed prompt length. When fixed tokens are set to 8192, the average speedup reaches $1.30\times$, compared to only $1.22\times$ at fixed = 4096. This is because the cache miss incurs higher overhead as the prefix length grows, and KVFlow's proactive cache management becomes increasingly beneficial.

**Optimization Breakdown.** We perform a breakdown analysis of KVFlow's two optimization components on top of SGLang w/ HiCache on H100. We observe that by enabling workflow-aware eviction alone on top of HiCache provides an average $1.11\times$ speedup. By further enabling overlapped prefetching optimization in addition to eviction, we can improve the speedup to $1.29\times$, demonstrating that both components contribute significantly.

**Overhead Analysis.** KVFlow adds negligible overhead. First, CPU-side priority assignment and prefetch scheduling overlap with GPU computation under SGLang's zero-overhead batch scheduler, and our profiling shows no exposed CPU cost. Second, prefetching does not interfere with decoding because token generation incurs no PCIe transfers, so KV movement overlaps with compute. Third, when prefetched KV is unused, neither PCIe bandwidth nor memory is wasted, as transfers run only during active compute, and buffers are overwritten in place.

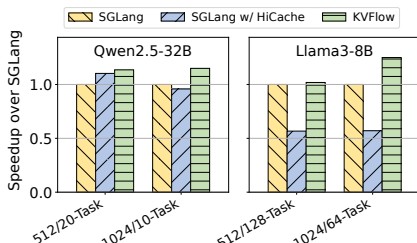 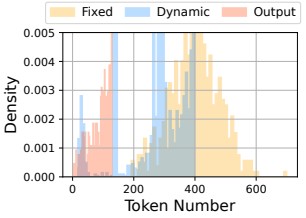 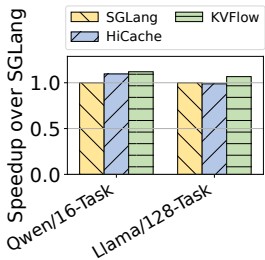

Figure 5: High-concurrency workflow performance comparison under different fixed-prompt/concurrency settings on an H100.

Figure 6: Token distribution for fixed, dynamic, and output parts across PEER-style workflows.

Figure 7: Speedup of KVFlow over SGLang and HiCache on PEER-style multi-agent applications.

## 4.2 High-Concurrency Workflow Performance

We further evaluate system performance under high concurrency by simultaneously launching multiple independent workflows on a single H100 GPU. These workflows are all sequential workflows (branches=1), and they are assumed to be non-interacting and non-sharing. As shown in Figure 5, we benchmark four configurations with models Qwen2.5-32B and Llama-3.1-8B, each labeled by the fixed prompt length per agent and the number of concurrent workflows. The dynamic and output token lengths are fixed at 256. For each setting, we choose a proper concurrency that the GPU can accommodate without exhausting memory for prefix caching. If concurrency is too high, all available memory is consumed by active requests, and the system can no longer maintain reusable prefix caches, placing it beyond the scope of our optimization.

**Results.** According to Figure 5, across all settings, KVFlow consistently outperforms both SGLang and HiCache, achieving an up to $1.25\times$ speedup. The performance improvement of KVFlow with 1024 fixed prompt tokens is higher than 512, as the cache miss overhead is higher. Notably, HiCache performs particularly poorly under high concurrency, even falling behind SGLang in multiple cases. For example, it only achieves $0.57\times$ performance of SGLang without CPU-based cache under 1024 fixed prompt tokens with 64 concurrent workflows. We suspect this is due to frequent cache misses triggering reactive load-back operations, which disrupt SGLang's schedule-compute pipeline. Additionally, due to the fragmented layout of KV storage in SGLang, the PCIe bandwidth cannot be fully utilized. While KVFlow also does not resolve the fragmentation issue, it achieves much better overlap of PCIe transfers and GPU compute through more reasonable evictions and proactive prefetching. As a result, it yields an up to $2.19\times$ performance gain over the naive LRU-based HiCache with reactive loading.

**Realistic Workflow Simulation.** To better reflect real-world deployment scenarios, we simulate agentic workflows based on the PEER [5] framework. In our setup, each workflow consists of four agents, instantiated using the workflow templates provided by PEER. For each agent, we sample a role and an instruction, and prompt an LLM to generate the agent's prompt. Due to the inherent randomness in LLM sampling, the generated prompts exhibit variability even when the roles and instructions are similar. Meanwhile, since all agents operate under the same application context, their prompts often share partially overlapping prefixes. This results in workflows that are both diverse and partially redundant, reflecting the common characteristics of real-world multi-agent applications.

We use the Financial QA dataset from PEER as the workflow input. The resulting workloads are moderate in scale, with agent prompts typically ranging from a few dozen to several hundred tokens. Figure 6 shows the distribution of fixed, dynamic, and output token lengths across all agents. Figure 7 presents the performance comparison between KVFlow, SGLang, and SGLang with HiCache. KVFlow achieves clear improvements over both SGLang and HiCache, resulting in up to $1.12\times$ and $1.08\times$ speedups. The results demonstrate the strong practical potential of KVFlow for multi-application serving in realistic deployment settings.

# 5 Related Work

**LLM Serving Optimizations.**    A broad line of work improves online LLM serving by optimizing request scheduling, including continuous batching (also known as iteration-level scheduling) [32], multi-level feedback queues to mitigate head-of-line blocking [33], quality-of-experience aware schedulers tailored to streaming scenarios [34], etc. Another set of efforts focus on KV cache management. vLLM proposes PagedAttention [11] to reduce memory fragmentation via paged storage of KV tensors, while SGLang introduces RadixAttention [12] to eliminate redundancy in prefix caching. Several works also target chatbot scenarios with specialized prefix caching strategies [16, 35]. InferCept [36] predicts tool calling durations and uses a cost model to decide whether to retain, swap, or discard the KV cache of intercepted requests. These optimizations are orthogonal to KVFlow, which focuses on workflows formed by multiple agents. While Autellix [37] and ParrotServe [38] explore request scheduling in agentic workflows, they do not consider prefix cache management, making their objectives complementary to ours.

**Agentic Workflow Frameworks**    Recent works have proposed diverse multi-agent frameworks [3, 4, 6, 7, 39, 40, 41] that organize agents into structured roles to collaboratively solve complex tasks. These frameworks provide built-in mechanisms for message passing and dependency construction between agents, convenient integration of tool usage and reasoning methods within agent actions, predefined abstractions for common agent roles and behaviors, and efficient multi-threaded execution to support concurrent agent collaboration. Some of these systems abstract the agentic workflow as a computation graph, where nodes represent LLM-invoking agents and edges denote control flow or message dependencies. This abstraction enables the application of graph-level transformations, such as edge pruning [42], operator insertion [8], or topology optimization [7, 10], to improve application correctness or quality. However, these frameworks remain focused on application-layer construction and rely on conventional LLM serving infrastructures to handle generation. In contrast, our work leverages the structure of agentic workflows to optimize the serving system itself, targeting backend efficiency under multi-agent execution workloads.

# 6 Conclusion

We present KVFlow, a workflow-aware KV cache management framework for optimizing LLM serving in agentic workflows. By abstracting agent execution as a Step Graph and computing each agent's steps-to-execution, KVFlow enables a principled eviction strategy that anticipates future usage. It further introduces a fully overlapped KV prefetching mechanism to proactively eliminate cache miss stalls. Our evaluations show that KVFlow significantly improves serving efficiency over existing systems in workflows with long prompts or high concurrency. While prior work on multi-agent systems has predominantly focused on designing frontend application logic and interaction protocols, KVFlow highlights the importance of workflow semantics in enabling system-level optimizations.

# 7 Acknowledgment

We sincerely thank the anonymous NeurIPS reviewers for their valuable feedback and insightful suggestions. We also appreciate the UCSD MLSys Group for their helpful comments. This work was supported in part by NSF grant 2124039, UC AI LEAP fund, and the Amazon research award.

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
