# OpenReview forum: "KVFlow: Efficient Prefix Caching for Accelerating LLM-Based Multi-Agent Workflows"
_NeurIPS.cc/2025/Conference — NeurIPS 2025 poster_

### Official Review · Reviewer_KtZF · 2025-06-30

**Clarity:** 2
**Significance:** 3
**Originality:** 3
**Rating:** 4
**Confidence:** 2

**Summary:**

This paper addresses the inefficiency issues when LLM serving for multi-agent workflows. The authors propose KVFlow, a workflow-aware approach to manage prefix caches  for agents with fixed prompts. Instead of the usual least-recently-used policy, KVFlow builds an “Agent Step Graph” to estimate when each agent will run next. Using this, the system evicts less-needed KV entries and even prefetches future entries in the background. The goal is to reduce cache misses and recomputation. Experiments (on synthetic Llama-3.1-8B and Qwen2.5-32B setups) show up to about 1.8×–2.2× speedup compared to a baseline GPU-only cache system.

**Questions:**

1. It is not clear how heavy the overhead of graph maintenance and computing priorities is. Can the authors provide more details on the runtime overhead of KVFlow itself?

2. The notion of “Agent Step Graph” assumes a known workflow in advance. How does KVFlow handle dynamic or conditional branches in a workflow that are not known beforehand?

**Ethical Concerns:**

["NO or VERY MINOR ethics concerns only"]

**Final Justification:**

Based on the rebuttal, I recommend that this paper be accepted.

**Paper Formatting Concerns:**

As I mentioned before, I think the title should be more concise.

**Quality:**

3

**Strengths And Weaknesses:**

**Strengths**

1. This is an important research problem. The paper clearly identifies a real bottleneck in agentic workflows, where naive LRU eviction causes frequent recomputation. Managing KV caches intelligently is an important problem for efficient LLM serving.

2. Using a workflow graph with steps-to-execution values for cache decisions seems novel. It is interesting to combine scheduling information with cache management. The idea of fine-grained eviction and fully overlapped prefetching is creative and practical.

3. The experimental results indicate noticeable speedups (up to ~2×) in relevant scenarios. The improvement is consistent across both single long workflows and many concurrent workflows, suggesting the method can benefit different settings. The quantitative gains are a strong point.

**Weaknesses**

1. The experiments use synthetic data (random token sequences) rather than real dialog or code tasks. Also, they compare mainly with one baseline (SGLang variants). There is little discussion of how the system performs under real-world workloads or against other existing optimizations.

2. The method is tailored for a specific “agentic” workflow setting with fixed prompts. The title mentions “Key-Value Routing” but the paper mostly discusses caching rather than any routing mechanism; this confusion reduces clarity.

---

> ### Author Rebuttal · Authors · 2025-07-30
>
> Thank you for your insightful comments and questions.
> We are happy to further improve our manuscript based on your suggestions.
>
> ### (W1) Workload and Baselines
>
> Thank you for the feedback. In addition to synthetic data, we evaluate KVFlow using dataset and agentic workflows derived from the PEER [3] paper in Section 4.2, which is a framework used widely in production. These workflows are semantically meaningful and simulate realistic agent coordination patterns.
>
> Workflows that are either fully deterministic or contain a limited number of branches are still common in current applications [1, 2, 3]. To further strengthen generalizability, we now include a 10-stage branching workflow, where agents are randomly selected at each stage from partially overlapping prompt groups. All configurations other than the branching pattern follow those in the paper.
>
> **Speedups over SGLang for branched workflows with Qwen2.5-32B on an H100**
> |Configuration|SGLang|HiCache|KVFlow|
> |-|-|-|-|
> |4096/32/32|1.000±0.016|1.099±0.024|**1.302±0.058**|
> |4096/256/256|1.000±0.012|1.001±0.017|**1.052±0.012**|
> |8192/32/32|1.000±0.001|0.754±0.014|**1.459±0.025**|
> |8192/256/256|1.000±0.003|0.924±0.008|**1.122±0.007**|
> |512/20-task|1.000±0.016|0.885±0.044|**1.225±0.021**|
> |1024/10-task|1.000±0.010|1.014±0.035|**1.140±0.007**|
>
> Regarding baselines, in addition to SGLang and its hierarchical cache extension, we now include the vLLM baseline, which organizes KV cache into blocks and adopts LRU-based block eviction policy. The results are shown in the following table. vLLM is slightly faster than SGLang in our experiments, but KVFlow still consistently outperforms both. Since KVFlow is built on top of SGLang with HiCache, comparing against SGLang better reflects the impact of our proposed optimizations. Our approach is general and can also be applied to vLLM or other LLM serving frameworks.
>
> **Speedups over SGLang for sequential workflows with Qwen2.5-32B on an H100**
> |Configuration|SGLang|vLLM|HiCache|KVFlow|
> |-|-|-|-|-|
> |4096/32/32|1.000±0.002|1.041±0.003|1.022±0.074|**1.389±0.045**|
> |4096/256/256|1.000±0.004|1.031±0.005|0.996±0.022|**1.043±0.010**|
> |8192/32/32|1.000±0.043|1.057±0.045|0.738±0.030|**1.198±0.030**|
> |8192/256/256|1.000±0.013|1.041±0.014|0.928±0.006|**1.138±0.003**|
> |512/20-task|1.000±0.029|1.091±0.033|0.950±0.048|**1.256±0.010**|
> |1024/10-task|1.000±0.020|1.046±0.021|0.983±0.043|**1.132±0.015**|
>
> > [1] Hong, Sirui, et al. "MetaGPT: Meta Programming for A Multi-Agent Collaborative Framework."\
> > [2] Qian, Chen, et al. "ChatDev: Communicative Agents for Software Development."\
> > [3] Wang, Yiying, et al. "PEER: Expertizing domain-specific tasks with a multi-agent framework and tuning methods."
>
> ### (W2) Title Clarification
>
> We apologize for the confusion. The term "KVFlow" in our title means **KV** cache management for agentic work**Flow**. We are open to refining the title to improve clarity.
>
> ### (Q1) Overhead Analysis
>
> Thanks for the question. The runtime CPU overhead introduced by KVFlow is minimal. First, even when the entire graph is very large, we only need to consider agents within a certain step (e.g., 5) of current activated agents. Second, SGLang’s zero-overhead batch scheduler [4] allows the scheduling logic on CPU to be overlapped with GPU execution, effectively hiding the associated runtime CPU cost. We observe negligible CPU overhead in our measurements. We will add this discussion in Sec 4 in the next version.
>
> > [4] SGLang v0.4: Zero-Overhead Batch Scheduler, Cache-Aware Load Balancer, Faster Structured Outputs.
>
> ### (Q2) Dynamic Workflow
>
> Thanks for the question. KVFlow is designed for workflows where the structure is known or can be inferred at runtime. KVFlow can handle dynamic workflows as long as the execution order of agents within a certain future window can be predicted at the current step. In such cases, KVFlow adaptively updates the steps-to-execution values at runtime and continues to make informed eviction and prefetching decisions. However, in cases where the workflow is highly unpredictable, KVFlow cannot anticipate future agent calls. In such scenarios, it falls back to HiCache’s LRU eviction and reactive cache loading. We will discuss this limitation in a dedicated section in our next version.

---

### Official Review · Reviewer_ojFE · 2025-07-01

**Clarity:** 3
**Significance:** 2
**Originality:** 2
**Rating:** 4
**Confidence:** 3

**Summary:**

This paper introduces KVFlow, a system for efficient KV (key-value) cache management in large language model (LLM)-based multi-agent workflows. The main motivation is that existing LLM serving frameworks, which often use an LRU policy for prefix KV caching, are not optimized for the structured and recurrent access patterns seen in agentic workflows. KVFlow addresses this by introducing a workflow-aware cache eviction policy grounded in an Agent Step Graph abstraction, enabling prediction-driven eviction at the KV node level. The framework also includes an overlapped prefetching mechanism to reduce CPU-GPU data transfer stalls, backed by status-aware scheduling. Experiments compare KVFlow with SGLang and SGLang's hierarchical cache extension under various agentic workflow loads, demonstrating notable speedups, especially for workloads with large static prompts and under high concurrency.

**Questions:**

How does KVFlow handle workflows with highly non-deterministic or user-interactive structures, where agent execution order may change unpredictably? Are there performance regressions or fallback mechanisms in such cases?

Can you provide more formal evaluation or ablation on the heuristic fixed/dynamic prompt segmentation’s impact? For example, is there any case where misclassification of prefix segments leads to inefficient cache usage or performance drops?

Have you assessed the overhead of maintaining and updating the Agent Step Graph and steps-to-execution metadata, especially in high-churn or large-scale deployments? Are there memory, CPU, or engineering challenges?

**Ethical Concerns:**

["NO or VERY MINOR ethics concerns only"]

**Final Justification:**

After considering the rebuttal, discussions, and clarifications from the authors, I find that several of my original concerns have been addressed, and the manuscript has improved in clarity and completeness.

Resolved Issues
Highly Non-Deterministic Workflows: Authors clarified KVFlow’s fallback behavior and added a limitation statement. They provided additional branched-workflow experiments showing consistent gains in moderately dynamic cases.
Fixed/Dynamic Prompt Detection: Provided both manual and heuristic strategies, discussed robustness to misclassification, and clarified fallback behavior without correctness loss.
Large-Scale Agent Step Graph Overhead: Explained that computation is localized to a small future window and can be hidden by GPU generation, citing SGLang’s scheduler.
Statistical Reporting: Standard deviations are now included for all performance results, demonstrating low variance and stable improvements.
Comparison to InferCept and vLLM: Clarified orthogonality with InferCept and included vLLM baseline results, showing KVFlow’s consistent advantage.

Remaining Considerations
While the approach handles moderately dynamic workflows well, performance in fully unstructured or highly unpredictable workflows remains inherently limited. This is acknowledged as a limitation.
Integration and evaluation with additional large-scale, real-world agentic workloads (beyond those in the paper) could further strengthen generality claims, but this is a minor point given the targeted scope.

Overall Assessment
The authors’ responses are thorough, supported by additional experiments and clear explanations. The work now clearly differentiates KVFlow’s contribution, shows robust performance across scenarios, and addresses methodological transparency. Remaining limitations are inherent to the problem scope and are acknowledged. I will maintain a positive score, as the rebuttal resolves key concerns and strengthens both the technical and empirical contributions.

**Limitations:**

yes

**Quality:**

3

**Strengths And Weaknesses:**

Strengths:
The paper addresses a clear inefficiency in LLM serving—LRU-based KV cache eviction poorly matches agentic workflows. KVFlow introduces a principled, workflow-aware eviction strategy using the Agent Step Graph abstraction, which models synchronization and branching dependencies (Fig. 3a). Its node-level priority assignment enables fine-grained cache management, particularly under prefix sharing scenarios (Fig. 3b). The proposed status-aware scheduling and overlapped prefetching effectively reduce CPU-GPU idle time (Fig. 4). Experiments across models, prompt sizes, and concurrency levels (Figs. 5–8) show up to 2.91× speedup over SGLang, validating performance claims. The work is reproducible in theory and relevant to real-world LLM use cases, especially in multi-agent and modular setups.

Weaknesses:
KVFlow is tailored for structured agentic workflows, limiting its applicability to unstructured or user-driven generation. Its reliance on accurate step graphs assumes robust frontend-backend integration, but fallback heuristics may not handle complex or ambiguous cases well. The paper lacks detailed evaluation on prefetching overhead and scalability in large or highly branching workflows. Experiments focus on synthetic or PEER-inspired patterns without broader real-world diversity, and statistical robustness is limited (e.g., no variance or significance analysis). Comparisons to relevant baselines like InferCept are missing. Some concepts, such as dynamic prompt detection, lack formal definitions.

---

> ### Author Rebuttal · Authors · 2025-07-30
>
> Thank you for your insightful comments and questions.
> We are happy to further improve our manuscript based on your suggestions.
>
> ### (Q1) Highly Non-Deterministic Workflows
>
> Thank you for the question. KVFlow is designed for structured agentic workflows where the execution path can be reasonably estimated.
> KVFlow can handle dynamic workflows as long as the execution order of agents within a certain future window can be predicted at the current step. In such cases, KVFlow adaptively updates the steps-to-execution values at runtime and continues to make informed eviction and prefetching decisions.
> In highly non-deterministic workflows, KVFlow gracefully falls back to default HiCache behavior by setting equal priority for all agent cache and disabling prefetching. We will clarify this in a distinct limitation section.
>
> Workflows that are either fully deterministic or contain a limited number of branches are still common in current applications [1, 2, 3]. We extend the evaluation with a 10-stage dynamic workflow with branches, and the results are shown in the belowing table, where all configurations other than the branching pattern follow those in the paper.
>
> **Speedups over SGLang for branched workflows with Qwen2.5-32B on an H100**
> |Configuration|SGLang|HiCache|KVFlow|
> |-|-|-|-|
> |4096/32/32|1.000±0.016|1.099±0.024|**1.302±0.058**|
> |4096/256/256|1.000±0.012|1.001±0.017|**1.052±0.012**|
> |8192/32/32|1.000±0.001|0.754±0.014|**1.459±0.025**|
> |8192/256/256|1.000±0.003|0.924±0.008|**1.122±0.007**|
> |512/20-task|1.000±0.016|0.885±0.044|**1.225±0.021**|
> |1024/10-task|1.000±0.010|1.014±0.035|**1.140±0.007**|
>
> > [1] Hong, Sirui, et al. "MetaGPT: Meta Programming for A Multi-Agent Collaborative Framework."\
> > [2] Qian, Chen, et al. "ChatDev: Communicative Agents for Software Development."\
> > [3] Wang, Yiying, et al. "PEER: Expertizing domain-specific tasks with a multi-agent framework and tuning methods."
>
>
> ### (Q2) Fixed/Dynamic Prompt Detection
>
> Thank you for the question. KVFlow allows users to manually specify the end position of fixed prompts based on application logic, or alternatively use a heuristic strategy. Specifically, we mark segments that are repeatedly reused by the same agent across invocations as fixed. Each agent can have multiple fixed prompts, forming a tree structure under shared prefixes. To avoid stale entries, KVFlow supports an adaptive mechanism that automatically removes fixed prompt nodes if they have not been reused beyond a threshold period.
>
> When the segmentation is partially incorrect, KVFlow still performs well without correctness issues. This case is analogous to workflows with a limited number of branches, as shown in the previous table, where KVFlow prefetches all candidate nodes and therefore achieves similar performance. In the extreme case where all fixed prompts are misclassified, KVFlow falls back to HiCache’s reactive cache loading or performs recomputation without performance degradation. We will add this discussion in Sec 3.3.
>
>
> ### (Q3) Large-Scale Agent Step Graph
>
> Thank you for the question. The overhead of maintaining the Agent Step Graph and updating steps-to-execution values is minimal. This is because we only need to consider agents within a certain step (e.g., 5) of current activated agents even when the entire graph is very large. The CPU overhead introduced by the scheduling of KVFlow can be hidden with GPU generation, thanks to SGLang's zero-overhead batch scheduler [4] design. We will add this dicussion in Sec 3.3.
>
> > [4] SGLang v0.4: Zero-Overhead Batch Scheduler, Cache-Aware Load Balancer, Faster Structured Outputs.
>
> ### Statistical Analysis
>
> Thank you for pointing this out. We now include standard deviations for key performance metrics across multiple runs. In all reported cases, KVFlow consistently outperforms baselines with low variance, confirming the stability of our results.
>
> **Speedups over SGLang for sequential workflows with Qwen2.5-32B on an H100**
> |Configuration|SGLang|vLLM|HiCache|KVFlow|
> |-|-|-|-|-|
> |4096/32/32|1.000±0.002|1.041±0.003|1.022±0.074|**1.389±0.045**|
> |4096/256/256|1.000±0.004|1.031±0.005|0.996±0.022|**1.043±0.010**|
> |8192/32/32|1.000±0.043|1.057±0.045|0.738±0.030|**1.198±0.030**|
> |8192/256/256|1.000±0.013|1.041±0.014|0.928±0.006|**1.138±0.003**|
> |512/20-task|1.000±0.029|1.091±0.033|0.950±0.048|**1.256±0.010**|
> |1024/10-task|1.000±0.020|1.046±0.021|0.983±0.043|**1.132±0.015**|
>
> ### InferCept Baseline
>
> Thank you for the comment. InferCept focuses on memory management during intra-agent interceptions such as tool calls, whereas KVFlow addresses prefix caching across agents by leveraging inter-agent execution patterns. These two approaches target different layers of the execution stack and are orthogonal in design. In fact, they could be combined: KVFlow’s eviction and prefetching mechanisms can manage inactive agents, while InferCept’s eviction policy can optimize memory usage within active, intercepted agents. We will clarify this distinction and potential complementarity in the related work section.
>
> Besides, we now include the vLLM baseline, which organizes KV cache into blocks and adopts LRU-based block eviction policy. Our experiments show that KVFlow outperforms it consistently. vLLM is slightly faster than SGLang in our experiments, but KVFlow still consistently outperforms both. Since KVFlow is built on top of SGLang with HiCache, comparing against SGLang better reflects the impact of our proposed optimizations. Our approach is general and can also be applied to vLLM or other LLM serving frameworks.

---

> > ### Comment · Reviewer_ojFE · 2025-08-06
> >
> > Thank you for your response. I appreciate the detailed explanations and the additional experiments you have provided.
> >
> > Your responses have successfully addressed all of my initial concerns. Specifically:
> >
> > Non-Deterministic Workflows: The new evaluation on dynamic workflow with branches provides convincing evidence of KVFlow's effectiveness even when the execution path is not fully deterministic.
> > Prompt Detection & Robustness: Your clarification on the manual and heuristic strategies for identifying fixed prompts, along with the adaptive mechanism for stale entries, is well-received.
> > Overhead: The reasoning that the overhead for the Agent Step Graph is minimal due to the local-window optimization and can be hidden by the GPU computation is logical and persuasive.
> > Statistical Analysis & Baselines: Thank you for including the standard deviations. Furthermore, the clarification of the distinction between KVFlow and InferCept is sound.
> >
> > My questions have been fully resolved. I retain my positive assessment of this paper.

---

### Official Review · Reviewer_Y4Pj · 2025-07-02

**Clarity:** 3
**Significance:** 3
**Originality:** 3
**Rating:** 4
**Confidence:** 3

**Summary:**

The paper presents KVFlow, an optimization framework designed to enhance prefix caching efficiency in large language model (LLM)-based multi-agent workflows. Existing LLM serving systems typically employ a least-recently-used (LRU) cache eviction policy, which often leads to unnecessary recomputation and degraded performance in workflows involving multiple agents with fixed prompts. To address this, KVFlow introduces two core innovations: (1) a workflow-aware eviction policy guided by an "Agent Step Graph," which accurately anticipates the execution order of agents, assigning cache eviction priorities based on their predicted temporal proximity; and (2) a proactive, fully overlapped prefetching mechanism that asynchronously transfers cached key-value (KV) tensors from CPU to GPU ahead of their anticipated reuse, thus minimizing latency. The authors implement and evaluate KVFlow within the SGLang framework, demonstrating significant latency reductions—achieving up to 1.83× speedup for individual workflows with large prompts and up to 2.19× improvement under high concurrency scenarios—compared to existing baseline approaches.

**Questions:**

1.  It is unclear how KVFlow handles dynamic workflows. The Agent Step Graph seems to be constructed from a known workflow structure. How would KVFlow deal with agents that are chosen at runtime (not just conditional branches) or workflows that evolve?
2.  The core KVFlow system is implemented on a modified SGLang backend. Will the authors release this code or detailed algorithms? Adding pseudo-code for key parts (priority assignment, eviction) would improve clarity. Also, explicitly state any hyperparameters (e.g. how far ahead prefetching looks) and whether KVFlow incurs extra overhead (CPU or memory).
3. The speedup results are compelling, but error bars or run-to-run variance are not shown. Please include standard deviations (or at least min/max) over multiple runs, especially for the main claims (Fig. 5–8).
4.  The paper emphasizes latency reduction, but what about trade-offs? For instance, does maintaining the Agent Step Graph and doing prefetching add CPU overhead or additional memory usage? Are there cases where proactive prefetching wastes bandwidth (e.g. if prefetched KV is not used)?

**Ethical Concerns:**

["NO or VERY MINOR ethics concerns only"]

**Final Justification:**

The major initial concerns regarding statistical rigor and methodological clarity were fully resolved, which significantly raised my confidence and assessment of quality and reproducibility. These aspects carry substantial weight and strongly influence my overall positive recommendation. Clarification regarding dynamic workflow support was sufficiently addressed, thus maintaining a high evaluation of generalizability and practicality. The remaining minor issue on quantitative overhead measurements slightly moderates my recommendation but does not substantially affect the paper’s overall merit or novelty.
Overall, the authors successfully addressed the most critical concerns, strongly supporting the recommended positive evaluation.

**Limitations:**

The paper does not include an explicit Limitations section, although the authors note they target “multi-agent workflows with limited GPU memory”. It would be valuable for them to acknowledge key assumptions: for example, KVFlow assumes knowledge of future agent calls and sufficient CPU memory for offloading. Discussing cases where these assumptions break (e.g. highly unpredictable agent invocation, memory-constrained environments without CPU caching) would give readers a more balanced view.

**Quality:**

3

**Strengths And Weaknesses:**

Strengths:
1. The paper identifies a clear performance issue in multi-agent LLM workflows, namely that naive Least-Recently-Used (LRU) prefix caching often evicts KV entries just before they are needed again. The authors propose a well-motivated solution, KVFlow, which leverages workflow semantics (via an “Agent Step Graph”) to predict agent invocation order and guide cache management. The methodology is technically sound.
2. The evaluation is thorough and well-structured.
3. The paper is generally well-written and organized.
4. The work shows practical impact: real deployments could serve more agentic workflows or use smaller GPUs at equal throughput.

Weaknesses:
1. Although the evaluation is extensive, it lacks statistical analysis. Reported speedups (e.g. 1.83×) appear to be averages, but no error bars or significance tests are shown. The paper does not discuss variance across runs, which is important for performance claims.
2.  The clarity of some methodological details could be improved. For example, the mechanism by which “steps-to-execution” priorities propagate through the shared prefix tree is described in prose, but no pseudocode or algorithm is provided. Readers may find it hard to reproduce the exact cache-priority logic.

---

> ### Author Rebuttal · Authors · 2025-07-30
>
> Thank you for your insightful comments and questions.
> We are happy to further improve our manuscript based on your suggestions.
>
> ### (W1) & (Q3) Statistical Analysis in Evaluation
>
> Thank you for the suggestion. We report the speedups with standard deviations in the table below. The results show that KVFlow delivers highly stable performance improvements. We will revise Section 4 to include these results.
>
> **Speedups over SGLang for sequential workflows with Qwen2.5-32B on an H100**
> |Configuration|SGLang|HiCache|KVFlow|
> |-|-|-|-|
> |4096/32/32|1.000±0.002|1.022±0.074|**1.389±0.045**|
> |4096/256/256|1.000±0.004|0.996±0.022|**1.043±0.010**|
> |8192/32/32|1.000±0.043|0.738±0.030|**1.198±0.030**|
> |8192/256/256|1.000±0.013|0.928±0.006|**1.138±0.003**|
> |512/20-task|1.000±0.029|0.950±0.048|**1.256±0.010**|
> |1024/10-task|1.000±0.020|0.983±0.043|**1.132±0.015**|
>
> ### (W2) & (Q2) Pseudocode
>
> Thank you for the suggestion. We will include pseudocode for both the eviction priority assignment and the eviction procedure in our next version. When a new agent request arrives with its Agent Step Graph (ASG) information, KVFlow updates the eviction priority of each cache node as follows:
>
> ```python
> def PriorityAssign:
>     for step, agent in ASG:
>         node = agent.last_fixed_prompt_node  # Last node of the agent's fixed prompt
>         while node != root:
>             node.counter[step] += 1
>             node.eviction_priority = min(node.eviction_priority, step)
>             node = node.parent
> ```
>
> When the system needs to evict cache nodes on the GPU to free memory (e.g., during prefill, decode, or prefetch), it proceeds from the leaf nodes in the cache tree based on their priorities:
>
> ```python
> def Evict:
>     leaves = get_leaf_nodes(tree_cache)
>     heapify(leaves) # Construct max heap based on eviction priority
>     while free_gpu_memory < required:
>         node = heappop(leaves)
>         free_gpu_memory += evict(node)
>         if node.parent becomes a leaf after eviction:
>             heappush(leaves, node.parent)
> ```
>
> For the prefetch distance hyperparameter, we use a value of 1 in our experiments, which is already good enough to hide data transfer. This value can be adjusted at runtime for different agents based on their workloads.
>
>
> ### (Q1) Dynamic Workflows
>
> Thank you for the comment. Our primary focus is on structured agentic workflows where future agent invocations can be estimated. KVFlow can handle dynamic workflows as long as the execution order of agents within a certain future window can be predicted at the current step. In such cases, KVFlow adaptively updates the steps-to-execution values at runtime and continues to make informed eviction and prefetching decisions.
>
> We extend the evaluation with a 10-stage dynamic workflow, where each stage randomly selects one agent from two branches with partially shared prefixes. The results are shown in the following table, where all configurations other than the branching pattern follow those in the paper.
>
> **Speedups over SGLang for branched workflows with Qwen2.5-32B on an H100**
> |Configuration|SGLang|HiCache|KVFlow|
> |-|-|-|-|
> |4096/32/32|1.000±0.016|1.099±0.024|**1.302±0.058**|
> |4096/256/256|1.000±0.012|1.001±0.017|**1.052±0.012**|
> |8192/32/32|1.000±0.001|0.754±0.014|**1.459±0.025**|
> |8192/256/256|1.000±0.003|0.924±0.008|**1.122±0.007**|
> |512/20-task|1.000±0.016|0.885±0.044|**1.225±0.021**|
> |1024/10-task|1.000±0.010|1.014±0.035|**1.140±0.007**|
>
> Workflows that are either fully deterministic or contain a limited number of branches are still common in current applications [1, 2, 3]. In highly unpredictable workflows where future execution cannot be inferred, KVFlow does not offer performance gains. In such cases, the system gracefully falls back to the default HiCache behavior by setting equal priority for all agent cache and disabling prefetching, without introducing additional overhead or correctness issues. We will include this discussion in a dedicated Limitations section.
>
> > [1] Hong, Sirui, et al. "MetaGPT: Meta Programming for A Multi-Agent Collaborative Framework."\
> > [2] Qian, Chen, et al. "ChatDev: Communicative Agents for Software Development."\
> > [3] Wang, Yiying, et al. "PEER: Expertizing domain-specific tasks with a multi-agent framework and tuning methods."
>
>
> ### (Q4) Overhead Analysis
>
> Thank you for the question. KVFlow introduces minimal runtime overhead. First, priority assignment and prefetch scheduling are handled on the CPU side. SGLang’s zero-overhead batch scheduler [4] allows these computations to be overlapped with GPU execution, effectively hiding the associated CPU cost. According to our profiling, we do not see CPU overhead that cannot be hidden by GPU computation. Second, prefetching does not interfere with LLM decoding, as token generation does not involve CPU–GPU data transfer over PCIe. This allows the transfer of prefetched KV to proceed in parallel with computation. Third, in cases where prefetched KV is ultimately not used, no PCIe bandwidth is wasted, since prefetching only takes place during active GPU computation. It also does not introduce additional memory overhead as the memory can be directly overwritten. If a cache miss occurs, the ongoing prefetch is safely preempted. While unused prefetching may incur additional energy consumption, our design prioritizes performance and responsiveness. We will clarify this discussion in Sec 4 in the next version.
>
>
> > [4] SGLang v0.4: Zero-Overhead Batch Scheduler, Cache-Aware Load Balancer, Faster Structured Outputs.

---

> > ### Comment · Reviewer_Y4Pj · 2025-08-07
> >
> > Thank you for your thoughtful and comprehensive response. Your detailed clarifications have effectively addressed my questions and reinforced my positive assessment of your work.

---

### Official Review · Reviewer_Gzk9 · 2025-07-05

**Clarity:** 2
**Significance:** 3
**Originality:** 3
**Rating:** 4
**Confidence:** 4

**Summary:**

This paper introduces KVFlow, a novel workflow-aware KV cache management framework designed to improve the serving efficiency of LLM based multi-agent workflows. Existing LLM systems often use a Least Recently Used (LRU) policy for KV cache eviction, which can lead to suboptimal performance due to frequent cache misses and recomputation in agentic workflows where agents' fixed prompts are repeatedly invoked. KVFlow addresses this by proposing two key techniques: a workflow-aware eviction policy and a fully overlapped KV prefetching mechanism. The paper demonstrates that KVFlow significantly reduces cache miss overhead.

**Questions:**

n/a

**Ethical Concerns:**

["NO or VERY MINOR ethics concerns only"]

**Quality:**

3

**Strengths And Weaknesses:**

- Strengths:

1. The paper identifies a crucial inefficiency in existing LLM serving systems, specifically the limitations of LRU-based KV cache eviction in the context of agentic workflows.
2. The introduction of the "Agent Step Graph" and the concept of "steps-to-execution" is a strong conceptual contribution. This allows for a more intelligent and anticipatory eviction policy that prioritizes KV caches based on their likelihood of future reuse within the workflow, moving beyond the limitations of simple LRU.
3. The paper describes the implementation of KVFlow as a prototype based on SGLang, highlighting its practical applicability and demonstrating how it extends existing systems.

- Weaknesses and Areas for Improvement:

1. While the evaluation includes sequential workflows and PEER-style simulations, exploring the performance of KVFlow on an even wider variety of agentic workflow structures (e.g., highly branching, deeply nested, or dynamic graph changes) could further solidify its generalizability. The current evaluation primarily focuses on scenarios where prefixes are likely to be reused, and a broader analysis of less predictable or more complex workflow patterns would be valuable to demonstrate robust performance across diverse real-world multi-agent applications.
2. The current implementation of KVFlow is presented as a prototype based on SGLang. While this demonstrates its feasibility, the paper could elaborate on the generalizability of KVFlow's design principles to other LLM inference frameworks (e.g., vLLM, DeepSpeed-MII, etc.).
3. A discussion on the architectural modifications or APIs required for plug-and-play integration into different existing multi-agent frameworks would strengthen the paper's impact and suggest broader applicability.
4. The paper states that "current systems typically evict KV caches using a Least Recently Used (LRU) policy," and then compares KVFlow to "SGLang with hierarchical radix cache." While SGLang's default eviction policy might implicitly be LRU or a variant, it would strengthen the comparison if the paper explicitly discussed and presented results against a clear, standalone LRU-based KV cache eviction mechanism as a direct baseline.
5. The paper would benefit from a more explicit ablation study to clearly demonstrate the individual contributions of KVFlow's two main components: the workflow-aware eviction policy and the fully overlapped KV prefetching mechanism.
6. It would be beneficial to discuss how KVFlow specifically handles or integrates with KV cache management strategies during the decoding phase.

---

> ### Author Rebuttal · Authors · 2025-07-30
>
> Thank you for your insightful comments and questions. We are happy to further improve our manuscript based on your suggestions.
>
> ### (W1) Evaluation Scope
>
> Thank you for the suggestion. We extend the evaluation with a new 10-stage workflow, where each stage randomly selects one agent from two branches with partially shared prefixes. Other configurations are following the paper. This setup introduces moderate unpredictability while still preserving opportunities for prefetching. As shown in the table below, for Qwen2.5-32B on an H100, KVFlow achieves up to 1.459$\times$ and 1.935$\times$ speedups over SGLang and SGLang w/ HiCache, demonstrating its effectiveness under branched execution.
> Workflows that are either fully deterministic or contain a limited number of branches are still common in current applications [1, 2, 3].
> While highly unpredictable workflows are outside our optimization scope, KVFlow can gracefully fall back to match that of HiCache by setting equal priority for all agent cache and disabling prefetching. We will add the results and discussion in Sec 4.
>
> > [1] Hong, Sirui, et al. "MetaGPT: Meta Programming for A Multi-Agent Collaborative Framework."\
> > [2] Qian, Chen, et al. "ChatDev: Communicative Agents for Software Development."\
> > [3] Wang, Yiying, et al. "PEER: Expertizing domain-specific tasks with a multi-agent framework and tuning methods."
>
> **Speedups over SGLang for branched workflows with Qwen2.5-32B on an H100.**
> |Configuration|SGLang|HiCache|KVFlow|
> |-|-|-|-|
> |4096/32/32|1.000±0.016|1.099±0.024|**1.302±0.058**|
> |4096/256/256|1.000±0.012|1.001±0.017|**1.052±0.012**|
> |8192/32/32|1.000±0.001|0.754±0.014|**1.459±0.025**|
> |8192/256/256|1.000±0.003|0.924±0.008|**1.122±0.007**|
> |512/20-task|1.000±0.016|0.885±0.044|**1.225±0.021**|
> |1024/10-task|1.000±0.010|1.014±0.035|**1.140±0.007**|
>
>
> ### (W2) Generalizability to Other Frameworks
>
> Thank you for the comment. KVFlow builds on general cache control and prefetching concepts and can be ported to other LLM serving systems. For instance, vLLM internally adopts automatic prefix caching and applies LRU at the block level. Our workflow-aware eviction logic can be applied at this block granularity by maintaining a priority score for each block, enabling anticipatory eviction and prefetching. We will add this discussion in Sec 3.3.
>
> ### (W3) API Modification
>
> KVFlow requires minimal extensions of traditional LLM call APIs. Specifically, we require (1) a client ID to distinguish requests from different workflows, and (2) the workflow information, including current activated agent name, and the steps-to-execution values for other agents. These inputs allow our runtime to assign priority scores for eviction and prefetching. We will add this illustration in Sec 3.3.
>
> ### (W4) Clarification on LRU Baseline
>
> Thank you for the comment, and we apologize for the confusion. Both SGLang and SGLang with HiCache adopt exact LRU eviction policies. The difference lies in how they handle evicted entries. SGLang stores KV cache only in GPU memory, so evicted entries must be recomputed when needed again. In contrast, HiCache backs up evicted entries to CPU memory, allowing them to be directly reloaded. We will revise Section 4.1 to clarify this distinction.
>
> ### (W5) Optimization Ablation
>
> Thank you for the suggestion. We performed a breakdown analysis of the two optimization components on top of SGLang with hierachical cache (HiCache). The table below compares the performance of using workflow-aware eviction alone with the combination of eviction and prefetching. Workflow-aware eviction alone provides an average speedup of 1.109$\times$ over HiCache, while enabling prefetching in addition to eviction further improves the speedup to 1.288$\times$.
>
> **Optimization breakdown for sequential workflows with Qwen2.5-32B on an H100**
> |Configuration|HiCache|+Workflow-Aware Eviction|+Prefetch|
> |-|-|-|-|
> |4096/32/32|1|1.236|1.358|
> |4096/256/256|1|1.037|1.047|
> |8192/32/32|1|1.217|1.623|
> |8192/256/256|1|1.000|1.223|
> |512/20-task|1|1.063|1.323|
> |1024/10-task|1|1.105|1.152|
>
> We also evaluated prefetching in isolation. However, in this case, the prefetched nodes frequently conflicted with those selected for eviction, leading to significant performance degradation. This indicates that prefetching must be combined with workflow-aware eviction in order to be effective. We will add these results and discussions in Sec 4.
>
>
>
> ### (W6) Cache Management Integration
>
> Thank you for the suggestion. To more clearly illustrate how KVFlow integrates with KV cache management, we will include pseudocode for both eviction priority assignment and the eviction procedure in the revised version. When a new agent request arrives with its Agent Step Graph (ASG) information, KVFlow updates the eviction priority of each cache node as follows:
>
> ```python
> def PriorityAssign:
>     for step, agent in ASG:
>         node = agent.last_fixed_prompt_node  # Last node of the agent's fixed prompt
>         while node != root:
>             node.counter[step] += 1
>             node.eviction_priority = min(node.eviction_priority, step)
>             node = node.parent
> ```
>
> When the system needs to evict cache nodes on the GPU to free memory (e.g., during prefill, decode, or prefetch), it proceeds from the leaf nodes in the cache tree based on their priorities:
>
> ```python
> def Evict:
>     leaves = get_leaf_nodes(tree_cache)
>     heapify(leaves) # Construct max heap based on eviction priority
>     while free_gpu_memory < required:
>         node = heappop(leaves)
>         free_gpu_memory += evict(node)
>         if node.parent becomes a leaf after eviction:
>             heappush(leaves, node.parent)
> ```

---

### Note · Authors · 2025-08-15

We sincerely thank the reviewers for their constructive feedback and insightful suggestions. We are encouraged that the novelty and practicality of KVFlow’s workflow-aware KV cache eviction and fully overlapped prefetching were recognized. During the rebuttal, we have made the following main improvements, which will be reflected in the final version:

1. Expanded Evaluation: Added experiments with dynamic workflows containing branching, showing that KVFlow still outperforms baselines in moderately dynamic settings. We also included comparisons with the vLLM baseline to demonstrate generality.
2. Methodology Clarification: Added pseudocode for both the eviction priority assignment and the eviction procedure to improve reproducibility.
3. Statistical Analysis: Reported standard deviations across multiple runs to convey performance stability more clearly.
4. Optimization Ablation: Conducted an ablation study breaking down the effects of workflow-aware eviction and overlapped prefetching, confirming that both contribute significantly to the observed speedups.
5. Overhead Discussion: Clarified that the scheduling overhead introduced by KVFlow is minimal and can be effectively hidden through SGLang’s zero-overhead batch scheduler.


We appreciate the reviewers’ time and engagement, and believe the updated paper addresses the raised concerns and presents a strong, well-rounded contribution to the ML and system communities.

---

### Decision · Program_Chairs · 2025-09-17

**Decision:**

Accept (poster)

**Comment:**

This paper proposes KVFlow, a workflow-aware KV cache management framework for accelerating LLM-based multi-agent workflows. It combines an agent step graph–based eviction strategy with a fully overlapped prefetching mechanism to significantly reduce cache misses and recomputation.

Before rebuttal, the main concerns centered on evaluation scope and generalizability (Reviewer Gzk9, Reviewer Y4Pj, Reviewer ojFE, Reviewer KtZF), methodological clarity (Reviewer Gzk9, Reviewer Y4Pj, Reviewer ojFE), and runtime overhead/handling of dynamic workflows (Reviewer Y4Pj, Reviewer ojFE, Reviewer KtZF).

Most reviewers acknowledge that rebuttal effectively addressed these issues by adding dynamic workflow experiments, clarifying LRU/vLLM baselines, and explaining overhead and fallback behavior.

Reviewers agreed that the paper identifies an important problem in LLM (Reviewer Gzk9, Reviewer KtZF), the proposed Agent Step Graph and workflow-aware eviction represent strong conceptual and practical contributions (Reviewer Gzk9, Reviewer Y4Pj, Reviewer ojFE), and the experimental results demonstrate consistent and significant speedups across scenarios with clear practical impact (Reviewer Y4Pj, Reviewer ojFE, Reviewer KtZF). All Reviewers vote positive to this paper Therefore, the AC recommend acceptance.